# The Impact of Mindfulness Interventions upon Visual Attention and Attentional Bias Towards Food Cues: A Systematic Review

**DOI:** 10.3390/nu17243885

**Published:** 2025-12-12

**Authors:** Ryan Duffy, Tuki Attuquayefio

**Affiliations:** 1School of Psychology, Western Sydney University, Sydney, NSW 2751, Australia; 2Translational Health Research Institute, Western Sydney University, Sydney, NSW 2751, Australia

**Keywords:** mindfulness, mindful eating, visual attention, attentional bias, mindfulness-based intervention, overeating, obesity, self-regulation

## Abstract

Background/Objectives: The so-called ‘Western diet’ characterised by the frequent consumption of high energy-dense (HED) food is linked with overeating, obesity, and an array of physiological and weight-related health complications. Attentional biases to HED food, which have been identified as a key mechanism promoting overeating, arise when reward-driven automatic processes impair the internal states responsible for regulating hunger and satiety. Emerging mindfulness-based interventions show promise in attenuating attentional biases by training controlled processes and enhancing the self-regulatory mechanisms required to override reward-driven automatic processing. Methods: Following PRISMA 2020 guidelines and PICOS strategy, this systematic review collates and synthesises current research on the impact of mindfulness interventions on visual attention and attentional bias to food cues in adults. Searches were conducted in Web of Science, PubMed, Scopus, Springer Nature, MEDLINE, Embase, and CINAHL in September 2025. Results: Findings obtained from six eligible studies were mixed indicating that mindfulness interventions significantly reduced attentional bias to HED, whereas other interventions indirectly enhanced self-regulatory systems such as hedonic hunger and craving without directly modifying attention. Additional findings highlight reductions in physiological reactivity, increased interoceptive awareness, and savouring. Conclusions: Overall findings suggest that mindfulness-based practices hold preliminary but promising potential to subdue attentional biases to HED food and disrupt unhealthy eating habits influenced by the Western diet. However, the limited number of studies, small sample sizes, methodological heterogeneity, and lack of mechanistic clarity indicate that such conclusions should be interpreted with caution. More robust and standardised research is warranted to determine whether mindfulness can produce durable, real-world behavioural change.

## 1. Introduction

The Western diet (WD) is characterised by the frequent consumption of highly palatable, ultra-processed, high energy-dense (HED) foods rich in saturated fats and sugars, and the reduced intake of nutrient-dense, fibre-rich foods that support metabolic and physiological health [1,2,3,4]. This pattern promotes excessive caloric intake and contributes directly to obesity and associated comorbidities [5]. Indeed, adherence to this diet leads to impaired memory [6,7,8,9,10], reduced sensitivity to hunger and fullness [7,8,9,10], and increased caloric consumption to feel full [8,10]. Moreover, those that consume more HED foods continue to want HED foods, even when full [7,8]. Within this dietary context, attentional biases toward HED foods—the preferential allocation of visual attention to high-calorie over low energy-dense (LED) foods have been identified as key cognitive mechanisms that promote overeating and weight gain [11,12]. Such biases can impair self-regulatory systems controlling hunger and satiety, increasing susceptibility to reward-driven food consumption [13,14].

### 1.1. Operationalising Visual Attention and Attentional Bias

Attentional bias is typically measured through reaction time (RT) or gaze-based paradigms assessing the preferential processing of food stimuli. The dot-probe task remains a standard approach, indexing bias through faster RTs toward probes replacing HED cues [15,16]. Similarly, the approach–avoidance task (AAT) measures implicit approach tendencies to food cues via joystick movements [17,18]. More recent eye-tracking methodologies provide continuous indices of visual attention such as fixation duration and gaze maintenance, offering ecologically valid measures of attentional bias to food cues in both laboratory and naturalistic settings [19,20].

### 1.2. Theoretical Frameworks Underpinning Attentional Bias

The dual-process model provides a conceptual framework explaining how attentional biases emerge through the imbalance between automatic, reward-driven processes (System 1) and controlled, reflective processes (System 2) [21]. System 1 operates rapidly and unconsciously, guiding behaviour through impulses, environmental cues, and learned associations. System 2 requires deliberate cognitive control to override such impulses. Continuous exposure to visually salient, heavily marketed HED foods reinforces reward associations that strengthen System 1 responsivity, reducing the inhibitory control exerted by System 2 [22,23]. Neuroimaging studies have demonstrated that HED food cues activate dopaminergic reward pathways, including the ventral striatum and orbitofrontal cortex, amplifying salience and attention capture [24,25]. Consequently, individuals high in food reward sensitivity show stronger attentional biases towards HED foods and greater consumption of discretionary snacks and fast foods [12,13].

### 1.3. Individual and Contextual Moderators

Attentional bias is influenced by several individual and contextual factors. Hunger enhances reward sensitivity and visual attention to HED foods by amplifying System 1 processing and diminishing satiety signalling [26,27]. Habitual exposure to the WD can further exacerbate these biases by impairing hippocampal function, which supports cognitive control of appetite [28,29]. Body mass index also moderates attentional bias, with individuals with overweight or obesity demonstrating stronger attentional capture by HED food cues and weaker self-regulatory control [11,30]. Moreover, clinical populations, particularly individuals with binge eating disorder, display heightened and prolonged attentional engagement with HED cues, driven by dysregulated self-regulation and increased reward sensitivity [31,32].

### 1.4. Mindfulness: Attentional Bias and Eating Behaviour Modification

Changing attentional biases towards HED foods may assist in reducing the frequent consumption of HED food and encourage healthy eating behaviours [33,34]. This is particularly important in the context of an obesogenic environment that facilitates the consumption of HED foods. Emerging mindfulness-based interventions show promise in attenuating attentional biases by training controlled processes and enhancing the self-regulatory mechanisms required to override reward-driven automatic processing. Mindful eating, an emerging construct of mindfulness, promotes healthier eating habits by increasing observation of internal states, focusing on hunger and satiety, weakening associations to external food cues, and supporting weight loss [35,36,37]. Mindful eating is defined as an awareness-based practice which involves facilitating attention to the present moment’s experiences of eating including the sensory and perceptual details of food, hunger and fullness cues, and thoughts and emotional states with a non-judgemental attitude [38]. Mindfulness interventions, particularly food- and eating-adapted mindfulness-based interventions including mindful eating (ME), or mindfulness-based eating awareness training (MB-EAT), aim to subdue visual attentional biases to HED foods through training controlled attention System 2 processing to interrupt and weaken the reward feedback loops and salience of HED food conditioned via automatic and impulsive System 1 responses to shift attention from HED to LED food, promoting healthier eating behaviours and food decision-making outcomes [23,35,36,39,40,41,42]. However, the literature regarding attentional bias modification towards HED food cues via ME and mindfulness-based interventions is limited. Given the role of attentional bias in both overweight and obesity and in the success of mindfulness interventions, there is a clear need to understand the relationship between mindfulness and attention. Specifically, the present systematic review synthesises current findings to determine the impact of mindfulness-based interventions on visual attention and attentional bias modification towards food cues within adult populations.

## 2. Materials and Methods

### 2.1. Eligibility Criteria

The following inclusion criteria were applied: experimental studies and randomised controlled trials (RCTs) relevant to mindful eating interventions, primary measure as visual attention or attentional bias towards food cues as the primary outcome, published in a peer-reviewed journal, in the English language, and have full text available. The following studies were excluded: cross-sectional, correlational or longitudinal design, systematic, narrative or scoping reviews, qualitative analysis, observational studies, studies with non-human participants, and studies with participants under the age of 18 years. The research question encompassed the PICOS criteria: Participants: Adults above the age of 18 across both sexes. Intervention: Mindfulness-based or mindful eating interventions. Comparison: Any control or non-intervention group allowing for comparison, including absence of mindfulness task, active, time- and expectancy-matched comparator (primary) cognitive training control; attentional bias modification (ABM); TAU/waitlist (acceptable secondary alternates for sensitivity analysis). Outcome: Visual attention or attentional bias towards food type. Study type: Experimental studies with random allocation to intervention or control groups.

### 2.2. Search Strategy

A systematic search was conducted across electronic databases Web of Science PubMed, Scopus, Springer Nature, MEDLINE, Embase, and CINAHL. An initial search focusing on visual attention and mindful eating interventions only was conducted on Web of Science on 6 September 2025, which retrieved insufficient results (n = 1). Therefore, the strategy was broadened to encompass attentional biases as an additional outcome and to include mindfulness as an umbrella term (see Appendix A for updated search strategy). An updated search was conducted on 10 September 2025, across Web of Science PubMed, Scopus, Springer Nature, MEDLINE, Embase, and CINAHL. The results of this search are described below. The review adhered to PRISMA guidelines [43]. Information was extracted from seven online databases: Web of Science, PubMed, Scopus, Springer Nature, MEDLINE, Embase, and CINAHL. The last search was conducted on 10 September 2025. The search strategy was generated from a number of keywords and key concepts in the field. The following terms were included: mindful eating, mindfulness-based eating awareness training, mindfulness, mindfulness intervention, visual attention, attentional bias, cognitive bias, information processing, attention control, eye-tracking, selective attention, eye movement, gaze direction, gaze pattern, dot-probe, visual search, stroop, fixation count, fixation duration, food, food cue, food stimuli, pictorial food, palatable food, high calorie food, unhealthy food, high energy density food, low calorie food, healthy food, and low energy density food. The full search strategy can be seen in Appendix A.

### 2.3. Study Screening

Studies identified via electronic sources were imported into Covidence systematic review software (Veritas Health Innovation, Melbourne, Australia) for title and abstract and full-text screening. Following removal of duplicates, 43 studies were eligible for screening. Two assessors (TA, RD) screened potential articles independently at every stage with 92.6% agreement. The studies were first screened based on their title and abstract and all conflicts were resolved by the collaborative understanding of the two researchers (TA, RD). Finally, 28 studies were full-text screened. A total of 5 peer-reviewed publications were identified and eligible for review from the electronic searches (see the PRISMA flowchart in Figure 1 for a full breakdown or Appendix A). One publication includes two experimental studies (i.e., there were six studies from five individual publications). Where both studies are relevant, we will refer to them simply as [44]; otherwise, we will specify the relevant study along with the citation (e.g., study 1, study 2). See Table 1 or Appendix A for data extraction.

## 3. Results

### 3.1. Study Design and Sample Size

Several study designs were used in this review; three (50%) were between-subject designs [44,45], one study (17%) used a quasi-design [46], one study (17%) employed within-subjects design [47], and one study (17%) implemented a mixed design [48]. The sample sizes for either intervention or control groups ranged from 10 to 50 [44,45,46,47,48].

### 3.2. Sex

The sex of the participants varied across the review. Two studies (33%) included 100% female participants [46,48]. Four studies (66%) featured both sexes [38,41,42]. No studies used males only.

### 3.3. Age Range

The mean age for participants ranged from 21.85 to 57.92 with all studies (100%) consisting of all adult participants [44,45,46,47,48]. Two studies (40%) did not report descriptive statistics [45,46], one study (17%) reported participant age as study inclusion criteria (30–50 years) [46], and one study (17%) presented age distribution in percentiles [45].

### 3.4. Mindfulness Interventions

In the testing conditions, all studies (100%) employed mindfulness interventions [44,45,46,47,48]. Four studies employed mindful eating interventions [Study 2 38,40,41,43]. Three studies (50%) employed multi-week interventions [45,46,48]. One study (17%) employed an eight-session mindfulness-based cognitive therapy (MBCT) programme combined with diet therapy, compared against diet therapy alone and a control [46]. One study compared eight weekly sessions of food-adapted mindfulness training (MT) featuring mindful eating principles and craving meditation, and a food-related attention bias modification training (ABMT), compared against a waitlist control [45]. One study (17%) implemented a 10-week mindfulness-oriented recovery enhancement (MORE) and personalised optimism with exercise recovery (POWER) programme compared to POWER alone [48]. Three studies (50%) implemented short scale single-session mindfulness interventions [38,41]. One study (17%) employed a brief mindfulness attention (MA) induction task compared against an immersed control [47]. One study (17%) implemented mindfulness meditation (MM) [Study 1 38] compared against a time- and expectancy-matched audiobook control. One study (17%) compared a mindfulness construal diary (MCD-R) [Study 2 38] against a time- and expectancy-matched audiobook control. Findings indicated that MBCT, ABMT, MA, and MORE significantly reduced attentional bias towards high-calorie palatable foods, while MM increased attention to low-calorie foods [Study 1 38,40–43]. In contrast, MT and MCD-R revealed limited or inconsistent effects on visual attentional biases [Study 2, 38,42].

### 3.5. Visual Attention and Attentional Bias Measures

Across the studies, attentional bias was most commonly assessed (33%) with dot-probe tasks [46,48]. One study (17%) employed a word- and image-based dot-probe contrasting food with neutral stimuli, with attentional bias indexed via reaction times (ms) to probe detection [46]. One study (17%) also utilised a dot-probe task, displaying palatable food images versus neutral household items at 50 ms (initial orienting) and 2000 ms (sustained attention), with bias measured through reaction times (ms) [48].

Three studies (50%) employed eye-tracking metrics [44,45]. Two studies (33%) utilised screen-based eye-tracking in free-viewing food image trials containing pairs of HED and LED foods for both studies, operationalised as mean and total fixation duration (ms) following a mindfulness meditation [Study 1, 38] and mindful construal diary [Study 2, 38]. One study (17%) used a modified dot-probe with eye-tracking containing HED and LED food images, recording initial fixation duration and sustained gaze duration (ms) [45].

One study (17%) employed an approach–avoidance task (AAT) which presented palatable food versus neutral food images overlaid with response cues, requiring joystick movements to approach or avoid [47]. Attentional bias was derived from the Food Attractiveness Index, calculated as RT differences (ms) between compatible and incompatible responses, reporting longer RTs in response to attractive food images compared to neutral food when an avoidance response was required compared to approach responses.

### 3.6. Studies with Additional Elements or Confounds

Several studies integrated additional elements beyond visual attention or attentional bias [44,45,46,47,48]. One study (17%) obtained physiological measures including BMI, systolic blood pressure, and diastolic blood pressure, reporting improvements in each alongside attentional bias reduction [46]. One study (17%) measured alpha-amylase activity and salivary volume as cephalic phase markers to food cue reactivity as well as self-reported food craving, and meta-awareness [47]. Two studies (33%) recorded state mindfulness, hunger, satiety, and BMI as potential confounds, reporting that both BMI and hunger did not significantly influence attentional outcomes [44]. One study (17%) obtained measures of hedonic hunger, mood, and binge eating frequency, including a bogus test of snack intake, revealing that reductions in attention correlated with reductions in binge eating [45]. One study (17%) recorded external eating behaviours, interoceptive awareness, savouring, and reward responsiveness using EMG to natural rewards, reporting that increases in savouring mediated reductions in food attentional bias [48]. Collectively, the pattern of these additional measures and elements suggests that mindfulness interventions not only directly influence attentional processing and attentional biases but indirectly act upon broader self-regulatory systems which support physiological arousal, emotional reactivity, and reward sensitivity towards food cues.

Most studies (60%) recruited healthy, normal-weight, non-clinical adult participant samples [44,46,47]. However, two studies (33%) recruited samples from clinical populations [45,48]. One study (17%) targeted adults with binge eating disorder (BED) or overweight and obesity, and psychiatric comorbidities [45], while another study (17%) targeted female cancer survivors with overweight or obesity [48]. These clinical populations may display stronger baseline attentional biases and difficulty disengaging, making it hard to disentangle intervention effects from group effects [6,25,26,27].

### 3.7. Risk of Bias

Critical appraisal was conducted on 11 September 2025 across the six included studies [49]. The full critical appraisal can be seen in Appendix A. The risk of bias across the included studies is variable and consistent. Selection bias: All studies (100%) used random allocation [44,45,46,47,48]. However, some studies were recruited via convenience sampling or provided limited detail on the procedures of randomisation and allocation when screening participants [46,47], therefore increasing the potential for systematic differences between groups and for baseline differences. Performance bias: The lack of participant and testing staff blinding across all studies raises the possibility of potential attentional or placebo effects [44,45,46,47,48]. Attrition bias: The small sample sizes in addition to variable dropout rates, often unreported, raise concerns regarding inconsistent and uncertain outcomes [44,45,48]. Reporting bias: Most studies emphasised the reporting of positive findings which inherently raises the potential for misleading conclusions [44,45,46,47,48]. The aforementioned factors raise a moderate-to-high risk of bias across the studies, and therefore findings should be considered as preliminary.

## 4. Discussion

### 4.1. Overview of Main Findings

This review synthesised experimental research on the impact of mindfulness interventions on visual attention and attentional bias towards food cues across six experimental studies. Overall, the findings showed that mindfulness practices can reduce automatic, reward-driven attention toward HED foods and enhance controlled, self-regulatory processes. Interventions such as MBCT, ABMT, MM, and MORE were effective in reducing attentional bias to palatable food cues and, in some cases, increased attention to LED foods [44,45,46,47,48]. From a theoretical standpoint, the findings here partially support the dual-process framework. Specifically, these outcomes align with the theoretical premise that strengthening controlled System 2 processes can inhibit impulsive, reward-based System 1 attentional capture [21,22,23].

Conversely, more general mindfulness interventions, such as the MCD-R and MT without a specific attentional component, yielded limited or inconsistent effects. Nonetheless, these approaches improved secondary self-regulatory mechanisms, including hedonic hunger, craving reduction, and emotional regulation [44,45]. Collectively, the findings suggest that targeted mindfulness interventions, particularly those adapted for eating behaviour, are more effective at modifying visual attention than generic mindfulness practices.

Beyond attentional outcomes, several studies demonstrated broader psychophysiological and emotional benefits, including reductions in blood pressure, improved interoceptive awareness, and increased savouring of natural rewards [47,48]. These results support the notion that mindfulness enhances awareness of internal states and modulates reward sensitivity, indirectly influencing attentional and behavioural responses to food cues.

### 4.2. Moderators of Attentional Bias and Mindfulness Effects

Several individual and contextual factors appear to moderate the impact of mindfulness on attentional bias. Hunger and physiological cue reactivity consistently heighten attentional bias toward HED foods [26,27], although few studies manipulated or statistically controlled for these variables. BMI and habitual diet also play important roles: individuals with overweight or obesity generally show greater reward responsivity and reduced inhibitory control [11,30]. However, this review found inconsistent moderation effects. BMI was unrelated to attentional outcomes in non-clinical samples [44,46], whereas clinical populations with obesity or BED displayed both heightened baseline attentional biases and greater reductions following mindfulness intervention [45,48]. Clinical populations such as those with BED or obesity showed stronger baseline biases [45,48], raising the possibility that mindfulness-based interventions may be most effective when reward-driven System 1 processes dominate and inhibitory control is compromised [23,25,26]. Identifying such moderators would help inform personalised or stratified intervention design. Indeed, a promising avenue for future studies is to investigate which individuals are most responsive to mindfulness interventions. Traits such as heightened hedonic hunger, emotion dysregulation, or binge eating symptoms may increase responsiveness to mindfulness-based strategies.

### 4.3. Clarifying Mechanisms of Change

While several studies reported reductions in attentional bias or improvements in self-regulatory mechanisms, the precise mechanisms driving these effects remain unclear. In particular, it is uncertain whether (a) reductions in attentional bias directly lead to behavioural change, or whether (b) improvements in interoceptive awareness and emotional regulation indirectly reduce impulsive eating. At present, studies often conflate these pathways; future research should seek to separate the influence of attentional control (System 2) from mechanisms such as craving reduction, savouring, or heightened interoceptive awareness. Based on dual-process theory, we propose that attentional bias may act as an initial cognitive inhibitor, while interoceptive awareness may facilitate long-term behavioural regulation, a hypothesis that future trials could test directly.

### 4.4. Duration and Intensity of Training

A key unanswered question is whether short interventions (e.g., single-session inductions) are capable of generating meaningful and lasting behavioural change. Given the nascent nature of this field, findings from the current review do not indicate whether attentional improvements persist over time. It is possible that a minimum “dose” of mindfulness practice may be required to strengthen controlled attentional processes sufficiently to override reward-based automatic responses. Future work should systematically test the threshold necessary for sustained change, using dose–response designs or longitudinal follow-ups.

### 4.5. Variability and Methodological Considerations

Substantial heterogeneity across study designs, participant samples, intervention formats, and attentional bias measures limits comparability and generalisability. Participant populations ranged from healthy adults to clinical groups with obesity or BED, who likely differ in baseline attentional control and reward sensitivity. Intervention formats varied in intensity and duration, from single-session mindfulness inductions to multi-week programmes (e.g., MBCT, MORE), while attentional bias was measured using diverse paradigms. Dot-probe and approach–avoidance tasks provide high experimental control but limited ecological validity, whereas eye-tracking yields richer, continuous measures of gaze behaviour at the expense of standardisation and control [17,50]. Given this variability, it is too difficult to determine which tasks, intervention types, designs, or populations are most amenable to mindfulness interventions.

### 4.6. Strengths, Limitations, and Future Directions

This review is the first to systematically synthesise experimental evidence examining the effects of mindfulness-based interventions on visual attention and attentional bias towards food cues. A key strength lies in its focus on an underexplored yet highly relevant domain, particularly as attentional mechanisms play a critical role in overeating and weight-related health outcomes. By assessing objective measures such as eye-tracking and reaction-time paradigms, as well as incorporating clinical populations, the review contributes novel insights into the biopsychological pathways underlying mindful eating practices and their potential influence on cognitive and behavioural regulation.

However, several important limitations must be acknowledged. Foremost, only six studies met the inclusion criteria, significantly limiting the strength and scope of conclusions. This small body of literature reflects not only methodological limitations but also the emerging and nascent nature of this research field, where standardisation and theoretical precision are still in development. Sample sizes across studies were small, with limited statistical power, and many studies lacked blinding and adequate follow-up periods. Additionally, the majority of experiments were conducted in controlled laboratory conditions using homogeneous samples, which may limit ecological validity and reduce confidence that outcomes would translate to real-world dietary behaviour.

Methodological heterogeneity further complicates synthesis. Considerable variability was present across study designs, intervention formats (e.g., brief inductions vs. multi-week programmes), attentional bias measures (dot-probe, eye-tracking, AAT), and population types (clinical vs. non-clinical). This variability made direct comparisons difficult and precluded meta-analytic evaluation. Furthermore, publication bias cannot be ruled out, as unpublished studies with null findings may remain unreported. Collectively, these factors indicate that findings should be interpreted as promising but preliminary rather than definitive.

Future research should prioritise greater methodological standardisation across mindfulness modalities, delivery formats, stimulus sets, and attentional measures. The adoption of consistent, validated food cue stimulus banks—particularly those distinguishing HED from LED foods—would enhance the comparability and ecological validity. Dose–response investigations are also warranted: it remains unclear whether single-session mindfulness inductions are sufficient to drive meaningful behavioural change, or whether a minimum threshold of practice intensity and duration (“minimal dose”) is required to produce lasting effects. Based on the present findings, we hypothesise that attentional bias modification may require sustained engagement with mindfulness practices in order to strengthen System 2 regulatory processes and override reward-driven, automatic System 1 responding.

Finally, the identification of individual difference moderators could help refine intervention targeting. Traits such as heightened hedonic hunger, binge eating symptoms, emotional dysregulation, or reduced interoceptive awareness may predict which individuals respond most strongly to mindfulness training. Evidence from the included clinical studies suggests that individuals with obesity or BED may display greater attenuation of attentional bias following intervention, possibly due to the dominance of reward-driven processes combined with weaker inhibitory control. Future research should therefore seek to stratify participants or use tailored intervention designs to examine whether specific psychological or clinical characteristics enhance treatment responsiveness. Lastly, a notable limitation across included studies was the predominance of female participants, which prevented the examination of sex-related differences. Future research should stratify samples by sex or recruit balanced groups to investigate whether attentional mechanisms or intervention responsiveness differ by sex, particularly given the evidence of sex-specific patterns in food-related reward sensitivity and disinhibition

Overall, while the current evidence is limited, it highlights fertile ground for advancing mechanistic understanding and refining intervention approaches. As this field develops, well-powered, ecologically valid trials using standardised protocols and longitudinal follow-up will be essential to determine whether mindfulness can meaningfully alter attentional processing and translate into real-world behavioural change and weight-related outcomes.

## 5. Conclusions

In summary, mindfulness interventions show preliminary but encouraging potential to reduce attentional biases toward HED food cues and enhance self-regulatory processes. However, due to the significant methodological limitations, small sample sizes, inconsistent mechanisms, and limited ecological validity, claims of sustained behavioural change or weight loss remain tentative. Future research should employ larger and more diverse samples, examine dose-dependent effects, clarify mechanisms of change, and evaluate real-world behavioural outcomes before strong causal inferences can be drawn.

## Figures and Tables

**Figure 1 nutrients-17-03885-f001:**
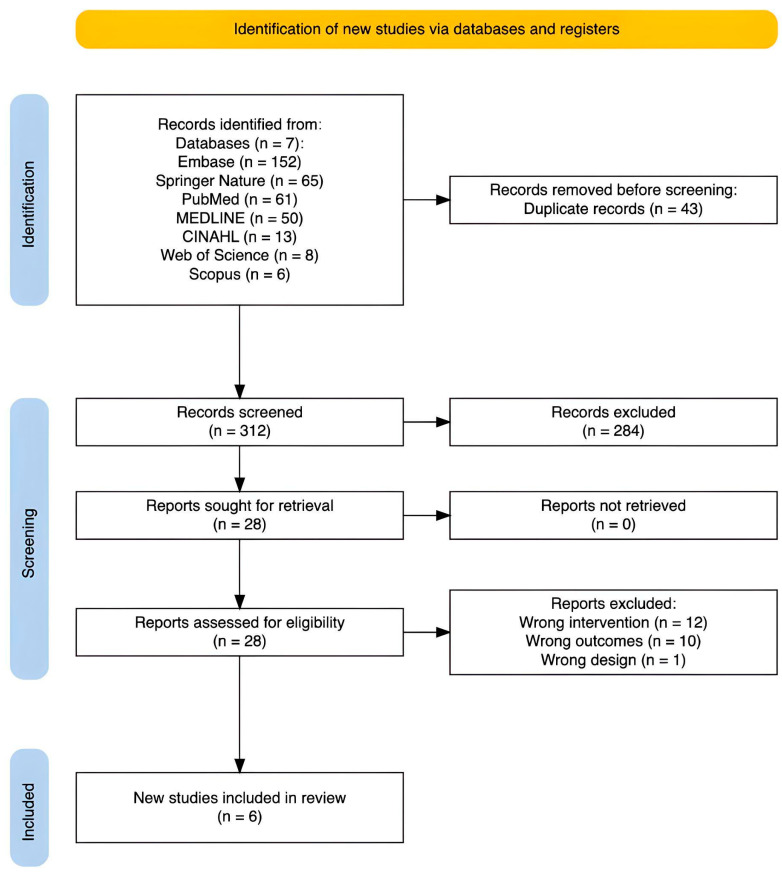
PRISMA flowchart for study search and screening.

**Table 1 nutrients-17-03885-t001:** Key findings for 29 articles analysed.

Author, Year, Country	Participant Population (N, Age, % Sex, BMI)	Clinical/Non-clinical	Study Design, Experimental Conditions	Blinding Reported	Mindfulness Intervention	Intervention Duration	Mechanism Assessed (Primary)	Measurement Tool(s)	Follow-Up Period	Key Findings	Ecological Validity (Lab/Real World)
Hussain et al. 2022 [44] England Study 1	N = 20, 75% female. Age 21.85 ± 3.18 BMI = 22.8 ± 5.14	Non-clinical	Between-subjects experiment (RCT)	Not reported	Mindfulness meditation (MM)	Single session	Visual attention and attentional bias	Screen-based eye- tracking (Tobii Pro X3-120) Metrics: Gaze duration, average fixation duration (ms), total fixation duration (ms).	None	MM increased state mindfulness and showed greater maintained attention toward LED food images vs. HED food images compared to control.	Lab
Hussain et al. 2022 [44] England Study 2	N = 44, 86.36% female. Age 23.61 ± 6.87 BMI = 24.44 ± 4.67	Non-clinical	Between-subjects experiment (RCT)	Not reported	Mindful construal diary (MCD-R)	Single session	State mindfulness and attentional bias	Screen-based eye-tracking (Tobii Pro X3-120) Metrics: Gaze directional bias, gaze duration bias, average fixation duration (ms), total fixation duration (ms).	None	The MCD increased state mindfulness but did not change maintained attentional bias to food relative to control. Participants displayed greater attentional bias towards HED food images compared to LED food images.	Lab
Mercado et al. 2023 [45] England	N = 45, 78% female. Age 31.6 years. BMI = 34.4 ± 6.57	Clinical	Between-subjects (RCT)	Not reported	Mindfulness training (MT) and attentional bias modification training (ABMT)	8 weeks	Attentional bias and hedonic hunger	Dot-probe task + eye-tracking (Tobii Pro Fusion remote eye-tracking) Metrics: RT differences (ms) + initial fixation duration bias (ms), fixation duration (ms), saccadic latency (ms).	4 weeks	Only ABMT significantly reduced attentional bias toward HED foods. MT increased ME and decreased hedonic hunger, emotional eating, and disinhibition.	Lab
Alamout et al. 2020 [46] Iran	N = 45, not reported, 100% female BMI = 27.28 ± 1.35	Non-clinical	Between-subjects design (RCT)	No	Mindfulness-based cognitive therapy (MBCT)	8 weeks	Attentional bias	Dot-probe task Metrics: RT differences (ms)	4 weeks	MBCT along with the conventional diet therapy was more effective in decreasing attentional bias towards food cues than the diet therapy alone.	Lab
Baquedano et al. 2017 [47] Chile	N = 50, 54% female. Age 23.9 ± 3.5, range 18–35	Non-clinical	Within-subjects design (RCT)	Not reported	Mindfulness attention (MA) induction task	Single session	Attentional bias and physiological response	Approach–avoidance task (AAT) Metrics: RT differences (ms)	None	MA showed a significant reduction in food approach bias to HED food compared to the immersed condition.	Lab
Thomas et al. 2019 [48] United States	N = 51, 100% female. Age 57.92 ± 10.04, BMI = 34.69 ± 7.39	Clinical	Mixed-subjects, stage I pilot (RCT)	No	Mindfulness-oriented recovery enhancement (MORE) and personalised optimism with exercise recovery (POWER)	10 weeks	Interoceptive awareness and attentional bias	Dot-probe task Metrics: Mean RT differences (ms) towards food vs. neutral image pairs (64 trial/block).	Post-test only	MORE + POWER decreased automatic FAB (50 ms cues) towards food images compared to POWER.	Lab–clinical hybrid

## Data Availability

The raw data and code supporting the conclusions of this article will be made available by the authors upon request.

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
