# Peer review of "The Impact of Mindfulness Interventions upon Visual Attention and Attentional Bias Towards Food Cues: A Systematic Review"

_nutrients, 2025, doi:10.3390/nu17243885_

Round 1
Reviewer 1 Report
Comments and Suggestions for Authors
There are various restrictions on this systematic review. First, just six research were included, which limits the quality of the body of evidence and makes individual study biases more likely to affect conclusions as a whole. Second, there was significant variation among studies in the tools used to measure visual attention and attentional bias, as well as in the kind, length, and delivery of mindfulness interventions. This unpredictability made it difficult to perform a quantitative synthesis or directly compare results. Third, it was challenging to draw definitive conclusions about the precise mechanisms of change because the results of different studies were inconsistent. Some showed improvements in attentional bias to high-energy dense foods, while others only showed improvements in related self-regulatory processes like craving or hedonic hunger.Fourth, there may be less trust in the dependability and longevity of stated benefits due to the methodological quality of the included studies, which are frequently characterized by small sample numbers, poor blinding, and brief follow-up periods. Fifth, even with thorough search methods, there is still a chance of publication bias because unpublished or null-result research might not have been found. Last but not least, the majority of the included research were carried out in controlled laboratory settings with rather homogeneous participant samples, which limited the findings' applicability to a variety of demographics and actual eating situations.
Author Response
Reviewer 1 Reply:
We thank Reviewer 1 for their time, constructive feedback, and thoughtful evaluation of our manuscript. We appreciate the acknowledgement that the English is clear and that the manuscript is generally well-presented. We have taken the reviewer’s concerns seriously and have made revisions throughout the manuscript to address the issues raised wherever possible. Below, we respond to each point in turn.
- “Only six studies were included, which limits the quality of the body of evidence and makes individual study biases more likely to affect conclusions.”
Response: We agree with the reviewer that the number of eligible studies was limited. This reflects the early and emerging nature of this research area, rather than a methodological choice. To address this point more transparently, we have expanded the Discussion sections, specifically the Limitations section (Section 4.6) to clearly acknowledge the small number of studies and the implications for strength of evidence. We now frame conclusions more cautiously, stating that the findings are preliminary rather than conclusive. We have included justification for the importance of synthesising early evidence, as systematic reviews are often essential in shaping future research agendas in novel fields.
- “There was significant variation between studies in the tools used and in the length/delivery of mindfulness interventions, which made direct comparison difficult.”
Response: We agree, and we have taken several steps to improve clarity and comparability. Table 1 has been revised to include columns for: intervention duration, type of mindfulness approach, attentional measurement tool used, ecological validity (lab vs clinical), and follow-up period. We now explicitly highlight methodological heterogeneity within the Discussion and Limitations sections. We also propose that standardisation of measurement tools and intervention protocols is a core recommendation for future research, directly addressing this issue.
- “It was challenging to draw precise conclusions on the mechanisms of change because the results were inconsistent.”
Response: We appreciate this observation and have now added a dedicated subsection in the Discussion titled “Clarifying Mechanisms of Change” (4.3)
where we distinguish between: attentional bias modification via controlled (System 2) processes, interoceptive/emotional regulation mechanisms, craving reduction and hedonic hunger outcomes We also now propose a testable hypothesis that attentional bias may act as an initial inhibitor for behavioural change, while interoceptive awareness may be responsible for long-term regulation, a conceptual advancement suggested for future studies.
- “The methodological quality of existing studies (small samples, poor blinding, short follow-up) limits confidence in long-term outcomes.”
Response: This point has been fully acknowledged in the revised Limitations section. We now clearly state that most studies used small samples and lacked blinding, emphasise that current findings should not yet be interpreted as evidence of sustained behavioural change and propose longitudinal and well-powered trials as a necessary step for establishing durable effects
- “Most studies were conducted in controlled laboratory settings with homogeneous samples, limiting generalisability.”
Response: This is an important observation. We have now suggested that future studies prioritise real-world settings, such as mealtime environments or mobile eye-tracking in naturalistic contexts and have acknowledged this limitation explicitly in both the Discussion and Limitations sections
Final Clarifying Statement: We respectfully note that many limitations identified by the reviewer are characteristic of a developing research field, rather than methodological flaws specific to our review. Systematic reviews at this early stage are often vital for identifying conceptual gaps, guiding methodological improvement, and shaping future research priorities, which we believe this manuscript now does more clearly and transparently.
We are grateful for the reviewer’s comments, which have helped strengthen the manuscript considerably.
Reviewer 2 Report
Comments and Suggestions for Authors
The manuscript addresses a timely and promising research area and provides value in several respects. Its strengths include examining a relatively underexplored topic while analyzing biopsychological mechanisms and employing objective measures, such as eye-tracking. Additional positives are the inclusion of clinical populations and the use of a dual-process theoretical framework, which convincingly justifies the relevance of the investigation. The manuscript is well-structured and comprehensive, with particularly strong background theory and methodology sections.
However, several limitations should be considered. The sample sizes in the included studies are very small, limiting statistical reliability. The considerable heterogeneity of the research methods, along with the limited ecological validity of laboratory tasks, reduces confidence that attentional changes would generalize to real-world eating situations. The samples were predominantly female, preventing analysis of sex differences. Moreover, some of the conclusions are stronger than what the data can fully support.
It would be useful to clarify which processes are primarily responsible for the effects of mindfulness. At present, it is unclear whether reductions in attentional bias lead to improved self-regulation, or whether enhanced interoceptive awareness reduces automatic, impulsive food choices. The manuscript often discusses these mechanisms together, but for drawing conclusions, it is important to specify which process is considered primary or dominant.
Regarding the duration of interventions, a key question is whether short, few-week trainings are sufficient to produce meaningful behavioral change, or whether a minimum intensity and practice duration are required to achieve lasting effects. The manuscript would benefit from at least proposing a hypothesis about the “minimal dose” needed for sustainable outcomes.
It would also be valuable to clarify whether specific personality or clinical characteristics , for example, stronger hedonic hunger or binge eating disorder , are associated with greater responsiveness to mindfulness interventions. Such information could inform the design of more targeted, personalized interventions.
In summary the findings are promising but preliminary and do not provide strong evidence that mindfulness interventions lead to sustained, real-world behavioral changes or meaningful weight loss. Most of the included studies are pilot in nature and methodologically heterogeneous, making strong causal inferences unwarranted. The authors should consider presenting their conclusions with appropriate caution and nuance.
Author Response
Reviewer 2 reply:
We sincerely thank Reviewer 2 for their thoughtful and constructive feedback. We greatly appreciate their recognition of the novelty of this research area, the strengths of the methodology, and the value of the dual-process theoretical approach. We have carefully considered all suggestions and have made revisions throughout the manuscript to clarify mechanistic pathways, improve transparency of limitations, and propose future research directions in line with the reviewer’s recommendations. Below, we address each comment in turn.
- “The sample sizes were very small, limiting statistical reliability, and laboratory-based tasks reduce ecological validity.”
Response: We agree with this concern and have now explicitly acknowledged these limitations within Section 4.6 (Strengths, Limitations, and Future Directions). We added a statement clarifying that small sample sizes and laboratory-based procedures limit generalisability, while also emphasising that this reflects the pilot nature and emerging status of the field. Additionally, we now recommend future studies employ larger, well-powered samples, real-world or naturalistic settings (e.g., mobile eye-tracking during meals), diverse participant groups to enhance ecological validity. Additionally, a new column for ecological validity has also been added to Table 1 to visually demonstrate these limitations.
- “The samples were predominantly female, preventing analysis of sex differences.”
Response: This point has now been clearly addressed within the Strengths, Limitations, and Future Directions (4.6) subsection of the Discussion. We highlight sex imbalance as a major limitation and propose that future research stratify or match samples by sex, enabling investigation of possible sex-dependent mechanisms. This aligns with the reviewer’s suggestion for more personalised intervention research. The sex imbalance is also a common artefact of the research recruitment process, and this systemic problem has been noted extensively elsewhere.
- “Some conclusions appear stronger than the data supports.”
Response: We appreciate this observation. Accordingly, The Conclusion section has been revised to present findings with greater caution, stating that results are promising but preliminary. Causal language has been softened throughout the manuscript. We now explicitly note that long-term effects and real-world behavioural change remain uncertain, and that current evidence does not yet support claims of sustained weight loss.
- “The manuscript should clarify which psychological process is primary - attentional bias reduction or interoceptive awareness.”
Response: We fully agree, and in response we have added a new subsection titled “Clarifying Mechanisms of Change” (4.3). This section distinguishes between, System 2 attentional control (e.g., AB modification, eye-tracking), interoceptive/emotional regulation mechanisms, and indirect pathways (e.g., craving reduction, savouring, hedonic hunger). We also propose a testable mechanistic hypothesis, “Attentional bias modification may act as an initial cognitive gateway for food-related behavioural change, while interoceptive awareness may drive long-term maintenance of self-regulatory control”. While we do offer a potential hypothesis regarding mechanism, we continue to highlight the preliminary nature of the field and our interpretation of the findings.
- “The manuscript should discuss whether a minimum duration of mindfulness training (‘minimal dose’) is required.”
Response: This suggestion has been implemented in a new subsection of the discussion titled “Duration and Intensity of Training” (4.4), where we now propose that future trials adopt dose–response designs to determine whether brief inductions are sufficient, or whether a minimum practice duration is necessary to produce lasting behavioural change. We explicitly refer to this as a ‘minimal dose hypothesis’, as recommended by the reviewer.
- “It would be valuable to clarify whether certain personality or clinical characteristics predict stronger responsiveness to mindfulness interventions.”
Response: We agree and have now expanded the “Moderators of Attentional Bias and Mindfulness Effects” (4.2) section to highlight potential candidate moderators, including, Hedonic hunger, Emotion regulation difficulties, Binge eating symptoms, and Reward sensitivity. We now suggest that stratified or personalised intervention designs should be explored in future research to determine for whom mindfulness interventions are most effective.
- “Findings are promising but preliminary and do not provide strong evidence of sustained real-world behavioural change.”
Response: We fully acknowledge this point and have revised both the Abstract and Conclusion to reflect that current findings are preliminary, long-term behavioural effects remain unclear, and the field requires more robust, ecologically valid research before strong causal claims can be made.
Final Statement: We are grateful to the reviewer for their insightful feedback. Their suggestions significantly enhanced the theoretical clarity, methodological transparency, and future relevance of the manuscript. We believe the revised version now more accurately reflects both the promise and limitations of this emerging research field.
Round 2
Reviewer 1 Report
Comments and Suggestions for Authors
Ok